# Plasmalogens: Free Radical Reactivity and Identification of Trans Isomers Relevant to Biological Membranes

**DOI:** 10.3390/biom13050730

**Published:** 2023-04-24

**Authors:** Carla Ferreri, Alessandra Ferocino, Gessica Batani, Chryssostomos Chatgilialoglu, Vanda Randi, Maria Vittoria Riontino, Fabrizio Vetica, Anna Sansone

**Affiliations:** 1Institute for Organic Synthesis and Photoreactivity (ISOF), National Research Council (CNR), Via P. Gobetti, 101, 40129 Bologna, Italy; alessandra.ferocino@kcl.ac.uk (A.F.); gessica.batani@isof.cnr.it (G.B.); chrys@isof.cnr.it (C.C.); fabrizio.vetica@uniroma1.it (F.V.);; 2Center for Advanced Technologies, Adam Mickiewicz University, 61-614 Poznan, Poland; 3Centro Regionale Sangue Regione Emilia Romagna (CRS-RER), Casa dei Donatori di Sangue, Via dell’Ospedale, 20, 40133 Bologna, Italy; vanda.randi@ausl.bologna.it (V.R.);

**Keywords:** fatty acids, cis-trans isomerization, free radicals, trans plasmalogens, erythrocyte membrane, plasmalogen analysis, lipidomics

## Abstract

Plasmalogens are membrane phospholipids with two fatty acid hydrocarbon chains linked to L-glycerol, one containing a characteristic cis-vinyl ether function and the other one being a polyunsaturated fatty acid (PUFA) residue linked through an acyl function. All double bonds in these structures display the cis geometrical configuration due to desaturase enzymatic activity and they are known to be involved in the peroxidation process, whereas the reactivity through cis-trans double bond isomerization has not yet been identified. Using 1-(1Z-octadecenyl)-2-arachidonoyl-*sn*-glycero-3-phosphocholine (C18 plasm-20:4 PC) as a representative molecule, we showed that the cis-trans isomerization can occur at both plasmalogen unsaturated moieties, and the product has characteristic analytical signatures useful for omics applications. Using plasmalogen-containing liposomes and red blood cell (RBC) ghosts under biomimetic Fenton-like conditions, in the presence or absence of thiols, peroxidation, and isomerization processes were found to occur with different reaction outcomes due to the particular liposome compositions. These results allow gaining a full scenario of plasmalogen reactivity under free radical conditions. Moreover, clarification of the plasmalogen reactivity under acidic and alkaline conditions was carried out, identifying the best protocol for RBC membrane fatty acid analysis due to their plasmalogen content of 15–20%. These results are important for lipidomic applications and for achieving a full scenario of radical stress in living organisms.

## 1. Introduction

Chemical and supramolecular properties of phospholipids are attracting interest for multidisciplinary applications, going from biologically relevant transformations [1,2,3], to nanobiotechnology, including liposome preparation and vesicle functions [4,5,6]. Plasmalogens (1-*O*-alk-1′-enyl 2-acyl glycerol phospholipids and glycolipids) are a class of phospholipids, the building blocks of cell membranes, with the general structure shown in Figure 1a. The vinyl-ether group in the *sn-1* position of L-glycerol is a peculiarity with respect to the ester function at both *sn*-1 and *sn*-2 positions, which characterizes the majority of membrane glycerophospholipids [7,8]. Recently, the orphan biosynthetic pathway relative to the formation of the cis vinyl function has been assigned to identify a delta-1 desaturase candidate gene [9,10]. Desaturase enzymes create fatty acid double bonds in a regio- and stereo-specifical manner, thus producing natural lipids as cis isomers with their biological effects depending upon such a configuration. In plasmalogens, the cis-vinyl ether function contributes to their behavior in biological structures; the cis geometry confers stronger intermolecular hydrogen bond interactions with contiguous phospholipids, influencing rigidity/fluidity, fission/fusion, and morphological transition [11].

The plasmalogen content reaches up to 15–20% of the total phospholipid mass in red blood cells (RBC), and increases in tissues like the heart (32–50%), brain (20–50%), and spermatozoa (55%) [12,13,14].

From the chemical reactivity point of view, both the cis-vinyl ether and the allylic functions are known to react under oxidative conditions, thus playing a “sacrificial” role toward a series of attacks arriving at the membrane compartment, as reported for singlet oxygen [15] and iron-induced peroxidation [16]. This distinctive susceptibility is translationally applied in clinics, to evaluate diseases suffering from oxidative stress, such as Down syndrome [17], neurodegenerative and cardiometabolic disorders [18,19,20], cancer [13], or physiological processes, such as aging [21]. Moreover, the vinyl ether function is susceptible to acid-catalyzed hydrolysis, forming a lysophospholipid derivative and liberating the fatty acid chain in *sn*-1 position as fatty aldehydes, in the form of dimethylacetal (DMA) when MeOH is present [22]. In the case of infectious diseases, such in vivo hydrolysis exerts toxicity effects, due to the reduction of membrane plasmalogens with health consequences [23]. It is worth underlining that fatty acid-DMAs are known to be formed under HCl-MeOH conditions used as the conventional protocol in the RBC membrane lipidome analyses, in order to convert the fatty acid residues of phospholipids into their corresponding fatty acid methyl esters (FAME) for GC evaluation. Indeed, the GC quantification of DMA can be used to estimate the RBC plasmalogen content [24,25].

The unique formation of trans-vinyl ether-containing plasmalogen was reported in the synthetic route to 1-(1E-octadecenyl)-2--oleoyl-*sn*-glycero-3-phosphocholine (Figure 1a with R_1_ as trans double bond and R_2_ as oleic acid, 9cis-C18:1) [26]. No information on the trans plasmalogen formation in biologically related conditions is available so far.

On the other hand, the conversion of natural cis fatty acids into their corresponding trans fatty acid (TFA) isomers is known to be an endogenous process promoted by the in vivo formation of RS^•^ species from thiols, which donate an H atom during the repair of free radical damages. The cis-trans isomerization reaction occurs by the mechanism shown in Figure 1c [27]. For polyunsaturated fatty acids (PUFA) it occurs randomly on one of the double bonds of these structures, and the product is a mixture of mono-trans isomers, as shown in Figure 1d for Ara with its four monotrans Ara isomers (mtAra) [27]. The synthetic library of mtAra obtained by laboratory procedures has been successfully used as a molecular standard to prove and follow up the endogenous Ara isomerization in cells [28,29], animals [30,31], and humans [32,33].

We were interested to work on plasmalogens to investigate their chemical behavior under free radical conditions, in particular concerning the formation of the corresponding trans configurations at both cis double bonds of the structure. We chose the representative molecule reported in Figure 1b, i.e., 1-(1Z-octadecenyl)-2-arachidonoyl-*sn*-glycero-3-phosphocholine (acronym: C18 plasm 20:4-PC), containing the cis vinyl ether function in a C18 hydrocarbon chain at the *sn*-1 position and the omega-6 PUFA arachidonic acid (Ara, 5c,8c,12c,15c-20:4) esterified at the *sn*-2 position of L-glycerol. For the first time, we clarified the plasmalogen reactivity involving these two reactive sites of plasmalogens. The trans-plasmalogen structure, synthetically obtained by photolysis, was isolated and characterized by ^1^H and ^13^C mono- and bi-dimensional NMR, to get new information on peculiar spectral signatures for analytical applications. A comparative study of oxidation and isomerization of this plasmalogen provided new insights into chemical reactivity, to complete the scenario of lipid classes, previously performed on glycerophospholipids and cardiolipins [27,28,34,35]. Using unilamellar vesicles and RBC ghosts, we carried out biomimetic studies on the supramolecular assembly created by plasmalogens in a mix with other phospholipids. We also deepened the hydrolytic behavior in basic and acidic conditions as part of robust chemical protocols for the analyses of fatty acids in liposome and RBC experiments [34]. Generally, we aimed at reaching a better comprehension of the plasmalogen molecular reactivity to translate chemical information into powerful diagnostic tools for precision medicine [36].

## 2. Materials and Equipment

C18 plasm 20:4-PC (1-(1Z-octadecenyl)-2-arachidonoyl-*sn*-glycero-3-phosphocholine) and DMA:18:0 (18:0 dimethyl acetal) were produced by Avanti Lipids (Birmingham, AL, USA) and purchased from Merck (Darmstadt, Germany); chloroform, methanol, diethyl ether, n-hexane, acetonitrile, benzene (HPLC grade) were purchased from J. T. Baker (Phillipsburg, NJ, USA); ferrous ammonium sulfate Fe(NH_4_)_2_(SO_4_)_2_ × 6H_2_O (Fe II AS) was from Carlo Erba, Milan, Italy; 2-mercaptoethanol, cis and trans FAME, Phosphate buffer, Hydrogen Peroxide, HCl/MeOH 0.5 M (1.5%; 1 mL/ampoule), BF_3_/MeOH 14%, Tetramethylsilane (TMS) were purchased from Merck (Darmstadt, Germany); POPC (1-palmitoyl-2-oleoyl-*sn*-glycero-3-phosphocholine) and SAPC (1-stearoyl-2-arachidonoyl-*sn*-glycero-3-phosphocholine) were purchased from Larodan (Solna, Sweden); benzene-d6 was purchased from CortecNet (Les Ulis, France); soybean lecithin was a gift from Lipoid GmbH (Ludwigshafen, Germany).

RBC was freshly isolated from anonymized blood samples of healthy volunteers received from the Department of Immunohematology and Transfusion Medicine (Metropolitan Area of Bologna, SIMT AMBO—Ethical Committee approval, #0083892).

NMR spectra were recorded at ambient temperature on a Varian 500 MHz spectrometer (Agilent, Cernusco sul Naviglio, Milan, Italy).

Hydrodynamic diameters of liposomes were measured using the dynamic light scattering (DLS) technique (Malvern Instruments Series NanoZS, Malvern Instruments, Malvern, UK) with a detection angle of 173°. All measurements were recorded at 25 °C.

Fatty acid methyl esters (FAME) were analyzed by comparison with authentic samples and chromatograms were examined and quantified by protocols that have been described previously [28,34]. 18:0 dimethyl acetal (DMA) was analyzed by GC (Agilent 6850, Agilent, Cernusco sul Naviglio, Milan, Italy), using the split mode (50:1) and a 60 m × 0.25 mm × 0.25 µm DB23 column (Agilent, Milan; Italy) with the same GC oven program and conditions used for FAME. DMA-18:0 was determined and quantified by its retention time and by multiple-point GC calibration curves of the standard reference (5 points) (LOQ: 0.0002 μg/mL; LOD: 0.0006 μg/mL; R^2^ = 0.9979). Photolysis was performed as already described [28,34].

Liposome reactions were performed in an incubating orbital shaker (Argolab, Ski 4, Carpi, Italy), keeping the temperature at 37 °C.

Ag TLC preparation, TLC glass plates were pre-treated with 5% AgNO_3_ solution in acetonitrile for 15 min and then dried at 120 °C for 1 h. Eluent for separation: CHCl_3_/MeOH/H_2_O 4.5:2.3:0.2. The formation of a mixture of trans isomers after 4 min of UV-photolysis was monitored by Ag-TLC, using the cis plasmalogen as reference.

## 3. Methods

### 3.1. Transesterification Procedures of C18 Plasm-20:4 PC, FAME, Soybean Lecithin and RBC

A stock solution of 5 mg/5 mL of C18 plasm 20:4-PC (99% pure) in CHCl_3_ was divided in aliquots of 0.25 mL in glass vials with Teflon cap (0.25 mg; 0.32 µmol) for each experiment. After the evaporation of CHCl_3_, transesterification reactions were performed in the following conditions in triplicates:

(A) alkaline conditions: addition of 0.5 M KOH in MeOH (0.5 mL) to the plasmalogen, stirring of the reaction for 10 min, and quenching with a saturated solution of NaCl (0.5 mL); the FAME were extracted with n-hexane (4 × 1 mL), the organic solvent was dried on anhydrous Na_2_SO_4_, evaporated to dryness, and the residue was dissolved in n-hexane (50 µL) and analyzed by GC (1 µL) [37];

(B) acidic condition 1: addition of HCl/MeOH 0.5M (1.5%) (1 mL; fresh opened single-dose ampoule) to the plasmalogen and stirring of the reaction for 2 h at a controlled temperature of 100 °C in a thermostatic block [38,39]. The cooled methanolic reaction mixture was then extracted with n-hexane (4 × 1 mL). The organic layers were collected and evaporated to dryness. The residue containing FAME was dissolved in n–hexane (50 µL) and analyzed by GC (1 µL);

(C) acidic condition 2: addition of BF_3_ in MeOH (14% wt/vol) (1mL) to the plasmalogen, stirring the reaction at a controlled temperature of 50 °C for 1 h in a thermostatic block [40]. After cooling at room temperature, the addition of 1 mL of a saturated solution of NaHCO_3_ and n-hexane (4 × 1 mL) was performed followed by vortex-mixing; the hexane layers containing FAME were separated, collected, dried over anhydrous Na_2_SO_4_ and evaporated to dryness. The residue containing FAME was dissolved in n–hexane (50 µL) and analyzed by GC (1 µL).

In all FAME extraction steps, 100 μL of a stock solution of internal standard 17:0 methyl ester (1 mg/mL) was added to calculate the FAME recovery yield.

Appendix A show the results of plasmalogen transformation.

Alkaline (0.5 M KOH in MeOH) and acidic (HCl/MeOH 0.5M) procedures with the addition of C17:0 FAME, as above described, were also applied to evaluate FAME recovery yield of soybean lecithin (1 mg). The indicated experiment was performed in triplicates and the results are shown in Appendix A.

The transesterification procedures were also tested with RBC. A single blood sample (40 mL), pooling 20 blood samples from different individuals, was created and 20 samples of 250 µL each (for a total of 5 mL) were taken to isolate RBC membranes and obtain membrane phospholipids, as described previously [32,33,41,42]. The 20 phospholipid extracts were divided into two groups to carry out alkaline (0.5 M KOH in MeOH) and acidic (HCl/MeOH 0.5M) transesterifications (n = 10 for each group). Only these two methodologies were chosen, excluding the acidic condition with BF_3_/MeOH that has been shown to give an extensive PUFA deterioration in the C18 Plasm-20:4 PC transformation. FAME was obtained and examined by GC analysis as above described.

### 3.2. Plasmalogen Cis-Trans Isomerization

Cis-trans isomerization was performed by applying the previously described procedure for cardiolipins [34]. C18 plasm 20:4-PC (20 mg, 0.025 mmol) was dissolved in benzene (2 mL); the solution (12.6 mM) was transferred to a micro-photochemical reactor, degassed under N_2_ for 20 min and, during the degassing, added with 0.5 equivalents of 2-mercaptoethanol in benzene (85 μL of a 146 mM stock solution; 6.3 mM). UV photo-irradiation was carried out for 4 min at a temperature of 22 ± 2 °C. Ag-TLC using CHCl_3_/MeOH/H_2_O 4.5:2.3:0.2 as eluent evidenced three fractions (Rf: 0.7, 0.8, 0.86 corresponding to trans plasmalogen isomers) in comparison with starting material (Appendix A). The reaction mixture was evaporated to dryness and left under vacuum for 6 h to remove the residual thiol reagent. The crude (19 mg) was dissolved in 0.3 mL of deuterated benzene to carry out NMR spectra (^1^H, ^13^C, 2D HSQC) and compare them with the same NMR experiments run on the starting material.

^1^H and ^13^C NMR spectra were performed on the commercially available C18 plasm 20:4-PC in deuterated benzene, a solvent chosen to avoid hydrolysis of the vinyl ether which occurs in deuterated chloroform (Appendix A). Identification and attribution of the resonances are described in SI. 2D HSQC NMR (Appendix A) was performed to assign proton and carbon resonances involved in double bonds, focusing specifically on those involved in vinyl ether moiety; Appendix A reports magnification of the regions corresponding to vinyl ether group and arachidonic acid alkenyl carbons. ^1^H and ^13^C NMR spectra in deuterated benzene were carried out on the mixture of cis/trans C18 plasm 20:4-PC after photolysis (Appendix A). The assignment of resonances in the mixture is described in SI. 2D HSQC NMR of cis/trans plasmalogen mixture was performed (Appendix A), to identify the vinyl ether group referred to as trans plasmalogen isomer (Appendix A). An aliquot of the plasmalogen isomer mixture (0.5 mg) was also treated with KOH/MeOH (as the transesterification procedure) to obtain the fatty acid methyl ester (FAME) of arachidonic acid (ARA) moiety present at *sn*-2 position. GC analysis under known conditions separated and identified the four mono-trans isomers of the ARA residue (Appendix A).

To quantitate the progressive formation of trans isomers, photo-irradiations were performed on C18 plasm-20:4 PC (5 mg, 0.0063 mmol, 0.5 mL benzene) in presence of 0.5 equivalent of 2-mercaptoethanol, for 1, 2.5 and 4 min. At each time point, as previously described, the solvent was evaporated under vacuum and the residue dissolved in 150 µL of benzene-d6 containing 7.4 mM tetramethylsilane (TMS) as internal standard.

### 3.3. Preparation of Liposomes

Liposome preparation followed previously published procedures [28,34]. Briefly, a mixture of 1-palmitoyl-2-oleoyl-*sn*-glycero-3-phosphocholine (POPC) (85%) and C18 plasm 20:4-PC (15%) was dissolved in chloroform, evaporated to dryness to obtain a thin lipid film and left under vacuum for 1 h to eliminate traces of solvent. Then 1.3 mL of tri-distilled water was added to reach a final lipid concentration of 10 mM and multilamellar vesicles (MLV) were formed by vigorous vortex stirring at 2200 rpm for 10 min. Large unilamellar vesicles were then prepared by extrusion technique (LUVET) using a filter of 200 nm diameter; the dimension was confirmed by Dynamic Light Scattering (DLS). The same procedure was applied for liposomes with 1-stearoyl-2-arachidonoyl-*sn*-glycero-3-phosphocholine (SAPC) 100%.

### 3.4. Preparation of RBC Ghosts

RBC was obtained from fresh EDTA-treated blood samples of healthy volunteers (see Appendix A) after separation from plasma as previously described [33,42]; briefly, 1 mL of whole blood was added with phosphate buffer (0.5 mL) and centrifuged at 5000 rpm for 5 min at 4 °C for two consecutive times. Plasma was removed and RBC lysis was then obtained by adding tridistilled H_2_O followed by subsequent centrifugation at 15,000 rpm per 15 min at 4 °C 4 times to obtain a clear supernatant, which was then discarded; the erythrocyte membrane pellets were collected and used to prepare RBC ghosts as multilamellar vesicles (10 mM lipids as MLV suspended in water) following the above-described procedure for liposomes.

### 3.5. Isolation of Plasmalogens from RBC Membranes

RBC membrane pellets, obtained following the protocol described in Section 3.4, were extracted with a 2:1 CHCl_3_/MeOH solution (4 times × 4 mL), according to the Folch method [43]. The organic layers were collected and dried over anhydrous Na_2_SO_4_, then evaporated under a vacuum to dryness. The lipid extracts were dissolved in 0.4 mL chloroform to isolate RBC plasmalogens by preparative TLC. The commercially available C18 plasm 20:4-PC was used as a standard reference. Eluent used for separation was CHCl_3_/MeOH/H_2_O 4.5:2.3:0.2 (adapted from ref [44]). The desired spot was scratched off, and silica gel was suspended in CHCl_3_/MeOH/H_2_O 4.5:2.3:0.2, stirred, and filtered; analytical TLC confirmed the purity of the plasmalogen fraction, estimated as 15% of the total lipid content.

### 3.6. Incubation Experiments under Oxidative Conditions

RBC ghosts or liposomes, prepared with SAPC and with 85:15 POPC/Plasmalogen mix (10 mM stock solution), were stored at 4 °C before use. Stock solutions of aqueous Fe(NH_4_)_2_(SO_4_)_2_ (1 mM), H_2_O_2_ (1 mM), and 2-mercaptoethanol (1 mM) were freshly prepared before incubations. To a 2 mL vial, the following reagents, at molarities similar to the biological ones, were added in sequence: Fe(NH_4_)_2_(SO_4_)_2_ × 6H_2_O (10 μM), H_2_O_2_ (100 μM), liposome (1mM) and thiol (10 µM). Control samples containing only liposomes (1mM), in the absence of iron salts and thiols, were used to estimate the incubation effect (as blank). In all cases, a final reaction volume of 0.5 mL was reached. The oxidation reactions were performed in open vessels, keeping an incubation at 37 °C in an orbital shaker for 15 h, as described for cardiolipins [34], followed by work-up (extraction, transesterification, and GC analysis) as described in Section 3.1 and Section 3.5.

### 3.7. Statistical Analysis

The results of the experiments were given as mean values ± SD and statistical analysis was performed using GraphPad Prism 8.0 software (GraphPad Software, Inc., San Diego, CA, USA). We used a non-parametric unpaired *t*-test two-tailed with 95% confidence interval.

## 4. Results and Discussion

### 4.1. Plasmalogen Cis-Trans Isomerization in Solution

The thiyl radical catalyzed isomerization of commercially available C18 plasm 20:4-PC (12.6 mM) was carried out in benzene by photolysis using 2-mercaptoethanol (6.3 mM) as the thiol [32,45]. We decided to perform the isomerization in an N_2_-saturated benzene solution since we observed that degradation of the starting material is easier in chloroform than in benzene, likely due to the acidity of the former solvent. Irradiation in the photochemical reactor equipped with a mercury lamp (λ = 250–260 nm) was carried out for 4 min monitoring the formation of trans isomers by Ag-TLC individuating two new spots at higher Rf (Appendix A). Indeed, the Ag-assisted technique assigns the highest Rf to the compound with more trans double bonds [46]. At this stage we tried to separate the two plasmalogen isomer fractions; however, we did not succeed due to the sensitivity to hydrolysis during the work-up. The cis/trans plasmalogen mixture was used for mono and bi-dimensional NMR experiments, in order to individuate characteristic resonances of the trans isomer configuration of the vinyl ether moiety as well as of the trans double bonds of the Ara moiety. Figure 2 summarizes the results, highlighting the NMR regions of ^1^H NMR (a) and ^13^C NMR (b) spectra corresponding to the vinyl ether function OCH=CH in the *cis* and trans configurations: in the starting material, the red line identified the connections between the proton signal of O*CH*=CH at ppm 6.22 with its carbon at 145.66 ppm; while the blue line correlates the proton *O*CH=*CH* at 4.55 ppm with the carbon at 107.07 ppm. The integration of the trans-assigned regions gave an estimation of trans isomer formation at 32% yield. The trans vinyl ether assignments are corroborated by analogous assignments in the case of the previously cited synthetic plasmalogen [26].

In Figure 2c,d, the traces of the cis/trans isomer mixture show the two vinyl ether proton signals and the corresponding carbon atom resonances (red and blue dotted lines) of the trans vinyl ether-containing plasmalogen isomer. In Figure 2c,d, ^1^H and ^13^C NMR analyses are dedicated to the arachidonic acid moiety of the plasmalogen structure. The spectral region corresponding to allylic and bis-allylic hydrogen atoms is focused with two signals at 1.97 and 2.78 ppm, respectively, related to the mono-trans isomer of plasmalogen in comparison to the natural cis plasmalogen.

In Figure 2d the ^13^C region corresponding to the C15 position of Ara residue is shown: the C-15 resonance of the cis isomer (130.34 ppm) is accompanied by four distinctive resonances at 130.90, 130.51, 130.26, 130.21 ppm, which belong to the mono-trans arachidonate isomers (mtAra) in position 14, 11, 8 and 5, respectively. This assignment is supported by previous data from the isomerization of 1-stearoyl-2-arachidonoyl-phosphatidyl choline (SAPC), and the four mono-trans isomers derived from the isomerization of Ara methyl ester (AraMe) [32,45].

^1^H NMR experiments were also performed on the mixture of cis/trans plasmalogen after photo-irradiation at 1, 2.5, and 4 min of C18 plasm-20:4 PC (5 mg, 0.00625 mmol, 0.5 mL benzene) in presence of 0.5 equivalent of 2-mercaptoethanol; the above-described characteristic proton signals were followed up by ^1^H NMR, at each time point of irradiation, dissolving in the crude reaction mixture in deuterated benzene 7.4 mM tetramethylsilane (TMS) as internal standard. This expedient allowed the calculation of the product yields, using the molarity of TMS standard to obtain those of cis/trans vinyl ether-containing compound, thus resulting in the yield of the trans plasmalogen (Table in Figure 3A). On the other hand, using the transesterification under alkaline conditions followed by quantitative GC analysis, the molarity of cis and mono-trans Ara isomers at the *sn*-2 position of plasmalogen was calculated (Appendix A). In Figure 3B the molarities expressed as relative quantitative percentages of the above-described analyses are reported in the function of the reaction time, showing the progressive formation of trans-plasmalogens (containing trans vinyl ether and mono-trans Ara isomers), reaching around 30% yield after 4 min photoirradiation, with 70% of the starting cis plasmalogen recovered in the mixture. Appendix A gives details of the follow-up of the cis and trans contents in this experiment.

It is evident from Figure 3 that the vinyl ether reactivity in cis-trans isomerization is similar to the reactivity of Ara moiety that contains four C–C double bonds. The mechanism of the cis-trans isomerization of C-C double bond in MUFA methyl esters by thiyl radicals has been studied in detail [47,48,49], as well as for methyl linoleate as a model for PUFA isomerization [50]. The reversible addition of thiyl radical to cis fatty acids converts them to trans fatty acids in a catalytic process (see Figure 1c) and the cycle is interrupted by exothermic allylic or bisallylic abstraction [49,50]. Kinetic data are available for all elementary steps as well as the cycle lengths for individual fatty acids. On this basis, the reason for the high reactivity of cis vinyl ether for the isomerization reaction (Figure 4) with respect to cis double bond in unsaturated fatty acids can be attributed at least to two factors [51]: (i) higher rate constant of RS^•^ addition due to the stabilization of adduct radical (see Figure 4), i.e., by the interaction of unpaired electron of thiyl radical with the lone pair electrons of oxygen, and (ii) the effect of the oxygen substituent increasing the pyramidality of the radical center and barriers to internal rotation about C^α^–C^β^ and C^α^–O, thus favoring conformational preference of intermediate radical towards the trans vinyl ether.

### 4.2. Transesterification Procedures for Plasmalogen-Containing Samples

As explained in Section 1, plasmalogens present a typical vinyl ether bond, which is resistant to strong bases and reducing/oxidizing agents and is easily cleaved by acids, leading to an aldehyde function further transformed to dimethyl acetal (DMA) when the reaction is performed in methanol [23,52,53,54]. We performed a precise evaluation of the best transesterification condition for releasing the fatty acid in the *sn*-2 position as the corresponding FAME. This step is very important to treat biological samples containing plasmalogens, such as in the case of red blood cell (RBC) membranes. Indeed, the occurrence of vinyl ether hydrolysis results in the formation of a lysophospholipid derivative, affecting the behavior of the PUFA residue in *sn*-2 [23]. We first compared treatments of the commercially available C18 plasm 20:4-PC using alkaline conditions at room temperature (rt) using 0.5 M KOH/MeOH, and under two acidic conditions: 0.5 M (1.5%) HCl/MeOH at 100 °C and 14% BF_3_/MeOH at 50 °C, as indicated in Section 3.1. Each experiment was performed in triplicate and results are reported in Appendix A. Only alkaline transesterification (0.5 M KOH/MeOH) transformed the arachidonic acid residue into its methyl ester (Ara-Me) in quantitative yield, as ascertained by GC quantitative analysis using 17:0 (as an internal standard in the extraction phase) and by calibration of standard Ara-Me via multiple points GC calibration curves, as previously described [34]. No formation of DMA18:0 was detected. The methanolic acidic condition at a high temperature (100 °C) gave Ara-Me 92% yield and parallel formation of DMA C18:0 (52% yield). The acidic treatment with 14% BF_3_/MeOH at 50 °C produced Ara-Me (58% yield) and DMA C18:0 (32.5% yield), showing that the two acidic conditions cause a consistent loss of information about the PUFA in the *sn*-2 position (Appendix A). Appendix A reports the GC chromatograms of the three transesterification reactions (A: KOH/MeOH at rt; B: HCl/MeOH at 100 °C; C: BF_3_/MeOH at 50 °C); under acidic reactions, the presence of Ara-Me, DMA 18:0 and 17:0 (reference standard) is detected, together with a degradation product of DMA 18:0 which is known to be due to additional hydrolytic processes [52,53,54]. The formation of DMA18:0 was also confirmed by GC-MS (Appendix A) with diagnostic fragmentations at *m/z* 75 and 283 [54]. These results clarify the only condition to treat plasmalogen-containing specimens, i.e., KOH/MeOH at room temperature, whereas the results of samples that contain plasmalogens under acidic conditions and high temperatures are inexact [24,25,53,54] since the PUFA residues present in the *sn*-2 position are not transformed quantitatively. Additionally, the use of acidic conditions, besides the high temperatures (50 and 100 °C) and longer reaction times required, needs fast manipulation and hermetic vials to avoid irritations and water interference; in the case of BF_3_ undesirable side reactions can also occur [55,56].

Alkaline and acidic transesterifications were also compared to examine the complexity of biological samples, such as soybean lecithin and RBC membrane pellet, taking into account that samples of biological origin and RBC often are reported under the condition of 0.5M HCl/MeOH at 100 °C [24,25,38,57]. In the lecithin experiment, performed in triplicate, the conversion of the linoleic acid residue into its methyl ester, calculated by calibration curve of the standard reference, was quantitative in basic condition (0.5 M KOH/MeOH at room temperature) and had a 93% yield in acidic condition (0.5 M HCl/MeOH, at 100 °C) (Appendix A). It is worth noting that plasmalogens are not present in soybean lecithin. In RBC membranes, where a natural content of plasmalogen is around 15–20% of the total mass of phospholipids (see Experimental Section 3.5) [12,13,14,58], a membrane pellet was prepared (n = 10) to carry out the two transesterification reactions of the same sample, followed by work-up and analysis, as described in Section 3.1.

The results of the FAME composition of the same RBC membrane pellet treated by the two transesterification reaction conditions evidenced clear and statistically relevant differences, as reported in Table 1. This can be expected from the well-known chemical reactivity of lipid structures which contains amide-, ether, and ester bonds to link the fatty acid moiety [58,59,60,61]. In the acidic condition, it was seen the enrichment of SFA residues, explained by the transformation of SFA residues belonging to amide-containing lipids, such as sphingolipids, which need a such strong protocol to be converted [59,60,61,62]. Acidic conversion is the most frequently used method for automated protocols of blood lipidomic analysis for epidemiological studies [62,63]; however, it must be clear to the analyst that in this case sphingomyelins are transformed (rich in SFA), whereas plasmalogens, as well as other acid-sensitive PUFA-containing lipids, are not stable. On the other hand, under basic conditions, plasmalogens can release the ester-linked fatty acid moieties, which are mostly PUFA residues, as FAME, thus saving this important information about the presence of essential fatty acids. In Table 1 the different PUFA contents in the two conditions can be compared. It is worth noting that the results are obtained quantitatively (µg/µL) by parallel 17:0 and PUFA calibration, and then transformed into relative quantitative percentages of each element in the fatty acid cluster (13 elements). In nutritional studies and in evaluating the oxidative effects of diseases, using RBC membranes, as well as sperm lipids and other specimens rich in plasmalogens, our results demonstrate the importance of laboratory protocol choice affecting the resulting outcome. Another important point is to use both SFA and PUFA standards for the fatty acid calibration step, to ascertain that work-up conditions do not affect labile structures.

### 4.3. Plasmalogen-Containing Liposomes as Model for Radical Stress Conditions

Radical stress conditions were then studied in plasmalogen-containing liposomes using the alkaline transesterification procedure to follow up the two expected processes: isomerization with the formation of mtAra isomers of plasmalogen and peroxidation reaction of the PUFA evaluated by the loss of Ara (Section 3.6). We previously evidenced that cis-trans isomerization can occur during lipid peroxidation [28,35]; however, plasmalogen reactivity by combined isomerization-oxidation processes was never reported. This is a missing point in the whole scenario of the sensitivity of this lipid class to oxidative conditions, including the protective role of vinyl ether bonds against PUFA peroxidation [15,16,26]. As extensively explained in previous work on lipid isomerization [27,28,29,30,31,32,34,35], we first discovered the harmful consequence of thiyl radicals formed from thiols for the cis lipid double bonds (Figure 4). The biological significance of this reaction relies on the fact that thiyl radicals are produced from sulfur-containing compounds, such as cysteine, glutathione, and H_2_S, as part of their antioxidant reactivity in cell metabolism [27,32]. We proved that trans fatty acid levels increase under biological stress conditions and nowadays these markers are considered in the scenario of pathological conditions [64]. The formation of free radicals from the antioxidant reaction is often under-evaluated in the whole scenario of biological damage repair. In our studies of the thiol involvement, we demonstrated that when the redox cycle of Fe^2+^/Fe^3+^ is involved, such as in the case of bleomycin complex, both reactive radical species OH^•^ and RS^•^ are produced and trigger the reactivity shown in Figure 5, causing these species both PUFA peroxidation together with MUFA and PUFA double bond isomerization, respectively [34,35].

We used this knowledge to carry out a biomimetic experiment with plasmalogens using a 1 mM liposome suspension, composed of an 85:15 ratio of POPC and C18 plasm 20:4-PC, thus keeping the 15% plasmalogen content usually present in RBC membranes. The natural content of oxygen in the medium (open air) was left in all reactions, i.e., no degassing operation was performed, as previously reported [34]. We added the Fenton reagent (10 µM Fe^2+^ salt and 100 µM H_2_O_2_) and incubated at 37 °C for 15 h, as the time frame used for the previously described cardiolipin experiment, with or without 10 µM amphiphilic 2-mercaptoethanol (Section 3.6).

In Table 2 the results of the two different suspensions as triplicates are reported, monitoring both peroxidation/isomerization of the Ara moiety of C18(plasm)20:4–PC (expected to give the four mono-trans isomers as depicted in Figure 1d) and isomerization for the oleic acid (9cis-18:1) moiety of POPC (i.e., expected to be transformed into elaidic acid, 9trans-C18:1 [27,28]). In the absence of thiol, no trans isomer was detected. The occurrence of peroxidation reaction was calculated from the recovery of Ara, quantitatively estimated using the calibration curve with Ara and the addition of an internal standard (C17:0 methyl ester). This is an indirect way to estimate Ara loss for peroxidation reaction, not using the direct measurement of peroxide formation [34,35]. In Table 2 the experiment without thiol showed no effect on the oleic acid moiety of POPC, whereas the Ara moiety of plasmalogen was reduced by 46%. In the presence of 10 μM 2-mercaptoethanol, as a source of thiyl radicals, the occurrence of both peroxidation and isomerization processes was evidenced for the oleic and arachidonic acids moieties, with the operative cycle shown in Figure 5.

Finally, the model of RBC ghosts was examined as an additional biomimetic model to estimate the plasmalogen contribution to the exposition of biological membranes to radical stress. They were obtained following known procedures described in Section 3.4 [33,43]. RBC ghosts are membranes containing 15% plasmalogen in a mix with other phospholipids [58,65] and allow evaluating the behavior of oleic acid (9cis-18:1), linoleic acid (9cis,12cis-18:2, PUFA omega-6) and Ara residues. In this way, we could reach a satisfactory overview of reactivity environments to evaluate how plasmalogens behave toward peroxidation/isomerization. The results are listed in Table 3.

The most representative PUFA are linoleic acid (26% of loss) and Ara (loss of 33%) after incubation at 37 °C for 15 h (Table 3). Compared with 85:15 POPC: C18 plasm 20:4-PC, where the Ara loss is 46% (Table 2), PUFA reactivity to oxidation can be considered almost equivalent. It is remarkable that in RBC ghosts, containing approximately 18% MUFA and 42% PUFA residues, including 18% Ara (see Table 1 and [58]), Ara loss resulted to be similar to the 15/85 plasmalogen/POPC liposomes. When thiol was added, the scenario of peroxidation and isomerization reactivity showed almost complete protection of PUFA moieties (linoleic and arachidonic acids yields = ca 95%) with the appearance of trans isomers in a 0.7–2.3%. The behavior of RBC ghosts was completely different from the 85:15 POPC: C18 plasm 20:4-PC liposomes, which still showed an Ara loss of 21%.

In order to compare the arachidonic acid reactivity in lipids without the vinyl ether function, we also compared the behavior of liposomes made of 100% 1-stearoyl-2-arachidonoyl-*sn*-glycero-3-phosphocholine (SAPC) under oxidative conditions. The consumption of Ara moiety under Fenton conditions was found to be very high (73%, see Appendix A), the highest among the liposome suspensions tested in this work. This can partly confirm the reported indication of oxidative protection given by the vinyl ether moiety of plasmalogens [15,16], at least reducing the PUFA loss comparing the data for liposomes containing 15:85 Plasmalogen/POPC, 100% SAPC, and RBC ghosts (Table 2, Table 3 and Appendix A). In this experiment, it must be underlined that plasmalogens are a mix of structures having choline and ethanolamine as the polar head. This structural diversity should be considered part of the variables that can lead to a different molecular reactivity, which can be deepened in further studies. It must be also added that although the influence of the cholesterol in membranes is not the target of this work, the cholesterol content in RBC ghosts [65] is known to protect membrane phospholipids from oxidative damage [66].

## 5. Conclusions

The present work expands knowledge on the plasmalogen reactivity toward oxidative conditions completing the scenario of biological membrane transformations. The phenomenon of membrane lipid oxidation involves lipid bilayer properties and interactions between the lipid bilayer and integral membrane proteins [67], as well as the type of membrane lipids triggering oxidative reactivity, such as in the case of ferroptosis [68]. We demonstrated for the first time that plasmalogens display an additional reactivity linked to the cis double bonds at the vinyl ether function and the polyunsaturated fatty acid residue. Geometrical isomerization and formation of trans plasmalogens occur in the presence of thiyl radicals, generated from thiols in connection with the redox cycle Fe^+2^/Fe^+3^, which is considered a double-edged process in the scenario of cell death [69]. The design of appropriate models to estimate the potential of oxidative processes and antioxidant effects in biological membranes must take into account also the molecular composition and the plasmalogen contribution in tissues. We furnished clear proof that lipid composition with its supramolecular organization plays a crucial role in the overall protection of membranes from degradative processes, which can vary from tissue to tissue and from their plasmalogen contents.

## Figures and Tables

**Figure 1 biomolecules-13-00730-f001:**
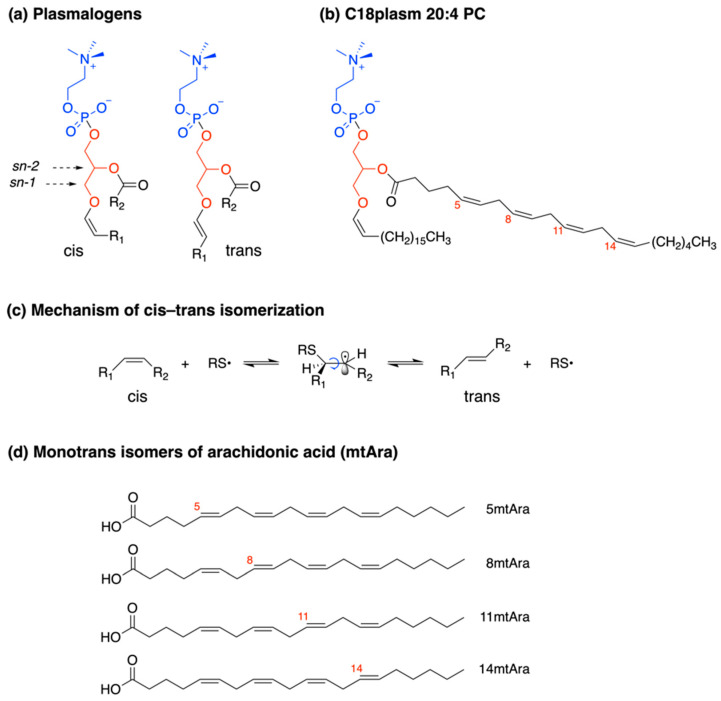
(**a**) General structure of plasmalogens with a fatty acid in position *sn*-2 and a vinyl ether-containing hydrocarbon chain in position *sn*-1. The natural structure has a cis-vinyl ether function. The trans-vinyl ether containing plasmalogen is a synthetically modified lipid. (**b**) 1-(1Z-octadecenyl)-2-arachidonoyl-*sn*-glycero-3-phosphocholine (acronym: C18 plasm 20:4-PC) with all cis double bonds. (**c**) Mechanism of thiyl radical catalyzed cis-trans double bond isomerization of unsaturated fatty acid moieties in lipids depicted as consecutive addition-elimination process. (**d**) The four monotrans isomers of arachidonic acid (mtAra) structures, as product mix obtained from the thiyl radical-catalyzed isomerization of Ara.

**Figure 2 biomolecules-13-00730-f002:**
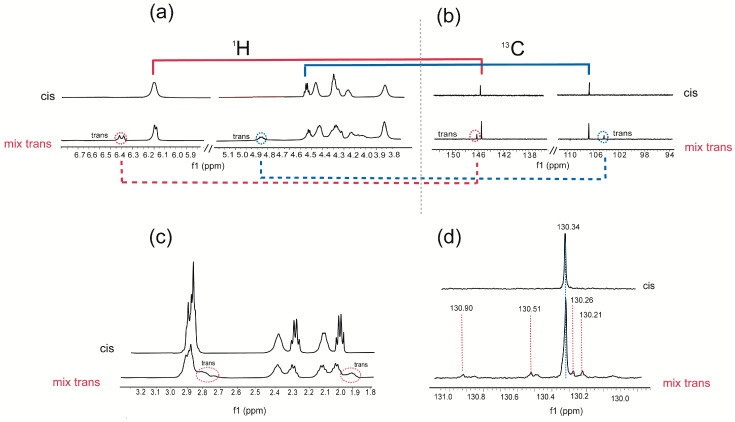
NMR spectra of the commercially available cis plasmalogen and the corresponding mixture of cis/ trans plasmalogen obtained after 4 min of UV photolysis; (**a**) ^1^H NMR and (**b**) ^13^C NMR regions relative to vinyl ether group; the red and blue lines give the H-C correlations of the OCH=CH group in cis configuration; red and blue dashed lines give the H-C correlations of the trans OCH=CH group; (**c**) region of bis-allylic and allylic proton signals; the red circle highlights the new signals at 2.78 and 1.97 ppm; (**d**) ^13^C NMR spectral region relative to C15 resonance of cis plasmalogen and the corresponding cis/trans plasmalogen isomers, where red dashed lines indicate the mtAra isomers: 14-trans (130.90 ppm), 11-trans (130.51 ppm), 8-trans (130.26 ppm) and 5-trans (130.21 ppm) isomers [32].

**Figure 3 biomolecules-13-00730-f003:**
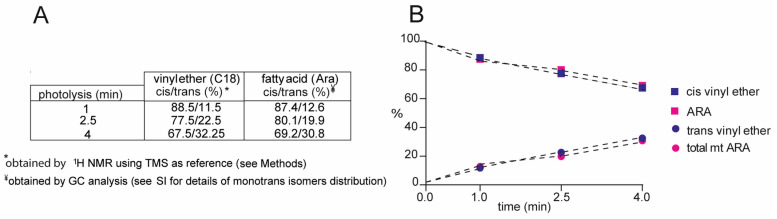
Follow-up of the isomerization of C18 plasm-20:4 PC (0.125 mM) performed in deuterated benzene in presence of 2-mercaptoethanol (0.5 equiv.) after 1, 2.5, and 4 min of photo-irradiation: (**A**) trans vinyl ether formation quantitatively determined by ^1^H NMR, using tetramethylsilane (TMS) molarity (7.4 mM) as a reference, and mt Ara isomers obtained by quantitative GC analysis after transesterification of the reaction mixture at each time point; (**B**) graphical representation of the values reported in Table A for the plasmalogen photoirradiation with the formation of trans isomers and decrease of the starting cis material. See details in Appendix A.

**Figure 4 biomolecules-13-00730-f004:**
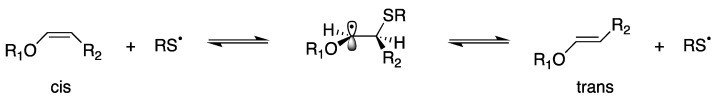
Proposed mechanism for the thiyl radical catalyzed cis-trans isomerization of vinyl ether moiety in plasmalogens.

**Figure 5 biomolecules-13-00730-f005:**
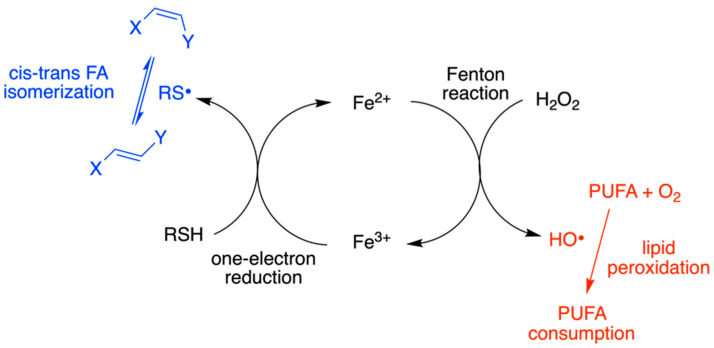
The isomerization and peroxidation processes occurring under Fenton-like conditions to double bond and PUFA moieties, respectively, in the presence of thiol RSH. The double bond can belong to the vinyl ether function or the unsaturated fatty acid chain.

**Table 1 biomolecules-13-00730-t001:** Comparison of fatty acid methyl ester (FAME) percentages (% rel. quant.) obtained from the same RBC membrane pellet transformed by alkaline (0.5 M KOH/MeOH; room temperature) and by acidic (0.5 M HCl/MeOH—100 °C) transesterification conditions (n = 10 repetitions each).

FAME ^1^	KOH/MeOH (% Rel. Quant.) Mean ± SD (n = 10)	HCl/MeOH (% Rel. Quant.) Mean ± SD (n = 10) ^2^
16:0	23.3 ± 0.6	34.0 ± 3.2 ***
9cis-16:1	0.24 ± 0.03	0.18 ± 0.05 **
18:0	16.0 ± 1.3	21.8 ± 1.3 ***
9trans-18:1	0.01 ± 0.01	0.03 ± 0.01 ***
9cis-18:1	17.0 ± 0.5	13.8 ± 1.0 ***
11cis-18:1	1.2 ± 0.2	1.0 ± 0.1 ***
9cis,12cis-18:2	12.1±0.6	9.2 ± 0.4 ***
20:3 omega-6, DGLA	2.3 ± 0.1	1.5 ± 0.3 ***
20:4 omega-6, Ara	18.8 ± 0.4	13.3 ± 1.4 ***
5mtAra	0.02 ± 0.01	0.03 ± 0.01 *
20:5 omega-3, EPA	0.8 ± 0.1	0.6 ± 0.3 *
22:5 omega-3, DPA	2.7 ± 0.5	1.7 ± 0.4 ***
22:6 omega-3, DHA	5.4 ± 0.1	3.0 ± 0.5 ***
SFA	39.3 ± 1.3	55.8 ± 3.3 ***
MUFA	18.5 ± 0.7	14.9 ± 0.9 ***
PUFA	42.1 ± 0.7	29.3 ± 2.6 ***
omega-6	33.2 ± 0.7	24.0 ± 1.8 ***
omega-3	8.9 ± 0.4	5.2 ± 0.9 ***
TOT TRANS	0.03 ± 0.01	0.05 ± 0.02 *

^1^ Identified by standard references and quantified as described in Materials and Methods. Values are obtained in µg/µL from the GC peak areas recognized and calibrated with standard references (corresponding to >97% of the total peaks of the chromatogram). Values are expressed in percentages relative to the sum of all the quantities of the recognized peaks (100%) ± Standard Deviation (S.D); ^2^ Statistical significance is estimated: * *p* < 0.043; ** *p* ≤ 0.0044; *** *p* ≤ 0.0005.

**Table 2 biomolecules-13-00730-t002:** Contents of 9cis-18:1 (oleic acid) and Ara and their corresponding trans isomers in 1 mM Plasmalogen/POPC liposome suspensions (15:85) in the presence of 10 µM Fe^2+^, 100 µM H_2_O_2_ incubated at 37 °C in the open air for 15 h with/without 10 μM 2-mercaptoethanol as a thiol ^1^.

Thiol (μM)	9cis-18:1 (%)	9trans-18:1 (%)	Ara (%)	Ara Loss (%)	mtAra (%)
0	99.9	-	54.0	46.0	n.d.
10	99.0	0.55	77.1	21.0	1.9

^1^ Values are means ± standard deviation of n = 3 experiments under the same conditions; they are expressed as a relative percentage (% rel. quant.) of each fatty acid isomer with respect to the starting cis fatty acid residue, estimated by GC analysis using calibration with appropriate standard references and C17:0 as internal standard. The sd is not indicated since all experiments had sd < 0.1; n.d. = not detectable.

**Table 3 biomolecules-13-00730-t003:** Yields (% rel. quant.) of 18:1 (oleic acid), 18:2 (linoleic acid), and Ara and their corresponding monotrans (mt) isomers obtained after transesterification of RBC ghosts (1 mM) incubated at 37 °C in the open air for 15 h in the presence of 10 µM Fe^2+^, 100 µM H_2_O_2_, with/without the addition of 10 μM 2-mercaptoethanol as a thiol.

Thiol (μM)	9cis-18:1	9trans-18:1	Ara	Ara Loss	mtAra	18:2	18:2 Loss	mt-18:2
0	99.0 ± 1.0	-	66.9 ± 1.1	33.1 ± 1.1	-	73.4 ± 2.1	26.5 ± 2.1	-
10	99.2 ± 0.1	0.7 ± 0.1	95.0 ± 0.2	2.3 ± 0.01	2.3 ± 0.2	96.1 ± 0.8	2.5 ± 0.6	1.4 ± 0.2

Values are means of n = 3 experiments under the same conditions expressed as a relative percentage (% rel. quant.) of each fatty acid isomer with respect to the starting cis fatty acid residue, estimated by GC analysis using calibration and 17:0 as internal standard; Yields of the three repetitions were found with errors <0.05%.

## Data Availability

The data presented in this study are available on request from the corresponding author. Data sets regarding humans subjects are not available due to privacy restrictions.

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
