# Peer review of "Plasmalogens: Free Radical Reactivity and Identification of Trans Isomers Relevant to Biological Membranes"

_biomolecules, 2023, doi:10.3390/biom13050730_

Round 1

Reviewer 1 Report

The authors have presented the plasmalogen reactivity under free radical conditions. They have carried out acidic and alkaline studies of plasmalogen reactivity and indicated the best method for the analysis of RBC membrane. Having carefully read through the script, some points for the attention of the authors are listed below.

Abstract

The abstract section is not clear enough, it needs to be rewritten. The authors should point the general topic being studied; the specific topic of the research; the central issues or statement of the problem to which your research relates; the main reason(s) and aims for your research; methods of research and/or analysis; your main findings; the significance or implications of your findings or arguments.

Introduction

How the current work gains novelty needs to be discussed in detail in the Introduction section.

Line 88 oxidative and free radical conditions? Maybe more appropriate would be in general free radical conditions.

Methods

Line 165 the authors should be more specific with the term brine.

Add reference for alkaline experiment.

Why the authors did not use all three transesterification methods for RBC? Please justify and correct it accordingly in the last paragraph of 3. Methods (line 190-196).

Line 278 Why did the authors perform 15h of incubation? Please, explain and if possible add the reference.

The authors mentioned that they use DLS for the measurement of size of the liposomes. What was the dimension of liposomes?

Discussion

The authors performed the experiment in the presence of thiol but did not discuss the biological significance and the sources of thiyl radicals in real systems.

Line 521 correct 2.5% to 2.3%

Conclusion

The conclusion section should be more concise and quantitative.

Reviewer 2 Report

The author has demonstrated, for the first time, that plasmalogens exhibit additional reactivity associated with the cis double bonds at the vinyl ether function and the polyunsaturated fatty acid residue. This topic is original in the field and addresses the chemical behavior of plasmalogens (specifically C18 plasm 20:4-PC) under oxidative and isomerizing conditions (mimicking conditions as in vivo). The results provide insight into the chemical transformations occurring in various lipid classes, such as glycerophospholipids and cardiolipins. The conclusions drawn are consistent with the evidence and arguments presented and effectively address the main question posed.

However, the study only employed one plasmalogen (C18 plams 20:4-PC) as the experimental material. To better understand the behavior of plasmalogens, it would be helpful to include Plasmalogen PE in the experiment. Additionally, while the amount of sn-2 vinyl ether plasmalogen is relatively low compared to sn-1 vinyl ether plasmalogen, it would still be beneficial to include one material of sn-2 vinyl ether plasmalogen in the study.

In conclusion, I recommend that this manuscript be published in Biomolecules once the authors have addressed our questions.
